# Univariate and Multivariate Statistical Analysis of Microbiome Data: An Overview

**Hani Aldirawi \* and Franceskrista G. Morales**

Department of Mathematics, California State University-San Bernardino, San Bernardino, CA 92407, USA
\* Correspondence: Hani.Aldirawi@csusb.edu

**Abstract:** Microbiome data is high dimensional, sparse, compositional, and over-dispersed. Therefore, modeling microbiome data is very challenging and it is an active research area. Microbiome analysis has become a progressing area of research as microorganisms constitute a large part of life. Since many methods of microbiome data analysis have been presented, this review summarizes the challenges, methods used, and the advantages and disadvantages of those methods, to serve as an updated guide for those in the field. This review also compared different methods of analysis to progress the development of newer methods.

**Keywords:** microbiome; OTU table; probabilistic model; regression analysis; zero-inflated model; hurdle model; longitudinal data





## 1. Introduction

The microbiome is a collection of complex microbial communities in the human body and other environments [1–3]. These communities have been found to constitute a large part of the human body systems and the environment. Developments in the study of the subject indicate that the state of the microbiome can determine the susceptibility to certain chronic conditions and diseases within various systems of the body including cardiovascular, gastrointestinal, respiratory, immune, and others. For example, within the gastrointestinal system, diet, medications, and environment have implications for the modulation of gastrointestinal health, which can be further investigated with the knowledge of the composition and function of the microbiome [4]. These implications suggest the importance of the microbiome in disease prevention and other health-related areas. This significance can be studied through microbiome–microbiome and drug/host–microbiome interactions, which could provide insight into the impacts on human health.

In microbiome–microbiome interactions, the environment plays a role in how different microbiomes interact with each other [5]. This encompasses the interaction of the human microbiome with microbiomes from the environment (e.g., pathogen development within the body). Host–microbiome interactions are how the microbiome and interactions within the microbiome affect the host. Drug–microbiome interactions focus on the interaction between drugs and the microbiome under study. Since the invention of antibiotics, the function of microorganisms in the body is more prone to the alteration of antibiotic resistance [6]. If these microorganisms are introduced to the environment, then more microorganisms will be able to inherit the resistance gene for certain antibiotics. Therefore, microbiome data analysis has implications for individual and community-wide health which can be applied through various methods. Microbiome analysis is a progressive area of study, and it is necessary to push experimental computational analysis and other methods in order to further investigate microbiomes and their interactions with various factors [7].

A basis of microbiome analysis is multi-omics. Most multi-omic studies focus on a separate analysis of each omics dataset without building a unified model. A major challenge in microbiome data analysis is the integration of multi-omics datasets [8,9]. This developing

method has future benefits in understanding the characterization of various environmental systems, industrial systems, and treatment processes [9]. The application of the multi-omics approach requires a combination of datasets from different omic groups to be analyzed. These omic groups—genome, proteome, transcriptome, and microbiome—allow for a new perception of the characterization and functions of microbial communities including gene expression, protein production, and community metabolism. Combining these techniques allows researchers to characterize the entire microbial community in greater depth by identifying other information such as gene expression, protein production, and community metabolism [9].

Data from multi-omics-based approaches can be analyzed further through post-data analysis, integrated data analysis, or model-based integration methods. The post-data analysis approach requires datasets to be analyzed individually so that connections between key features can be made afterwards. Integrated data analysis requires specialized tools to combine the datasets being analyzed so similarities can be identified statistically instead of being interpreted by humans [9]. Model-based integration methods require the system of study to be well-defined in order to compare new findings to the model. Since many of the systems of multi-omics are not fully characterized, this method is limited to the systems that are already characterized and defined [9].

Microbial analysis can have significant implications for areas beyond human health. For example, Mohan et al.'s (2014) multi-omics study used metagenomic and metabolomic techniques to analyze the water of hydraulic fracturing wastewater during fracking through metagenomic and metabolomic techniques [10]. Data from microbial communities in both sources of water were collected, tested, and analyzed. The results indicated that microbiomes in potable water have increased genetic ability to handle stress which has implications for biofilm control and microbial-influenced corrosion control. The implications of using fossil fuels have also been evident from a multi-omics study that analyzed the production of biofuels as an alternative [11]. The diatom Thalassiosira pseudonana was used to promote a high lipid yield, in order to progress the development of biofuels. Results from this study brought new perspectives to the development of biofuels, which emphasizes the importance of microbiome analysis beyond human health.

In this paper, we introduce various methods of modeling microbiome data. In Section 2, we discuss the microbiome data representation and some modeling challenges related to microbiome data. In Section 3, we discuss the types of models for sequence read counts from a single microbiome feature including probabilistic models, regression analysis, and longitudinal data analysis. In Section 4, methods of multivariate microbiome analysis are reviewed. We discuss the microbiome–microbiome interaction modeling strategies, host/drug–microbiome interaction, regression analysis, and some well-known longitudinal data models for multivariate count response. We conclude in Section 5.

## 2. Microbiome Data Representation and Modeling Challenges

The microbiome data are often sparse with a high proportion of zero values. These zeros have two possible sources. First, some species are truly never represented because they do not exist (biological zeros). Secondly, some species exist but are not detected as a result of insufficient sequence depth or inefficiencies of the technological processes (non-biological zeros) [12,13].

Microbiome data are high dimensional data, in general, as the number of species is greater than the number of samples in many situations. The microbiome data are over-dispersed (i.e, the variance is much higher than the mean). When the data are sparse with a high proportion of zeros, the distribution of the Operational Taxonomic Unit (OTU) is skewed [14]. Therefore, the OTUs cannot be correctly analyzed using standard baseline distributions such as Poisson and negative binomial distributions. The below Figure 1 is an example that shows the sparsity and right skewness of the feature distribution.

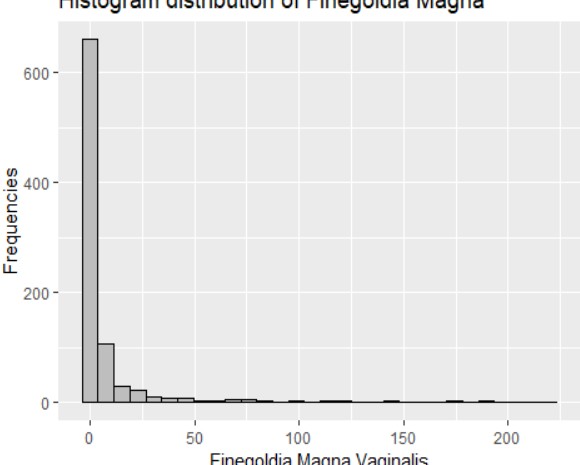

**Figure 1.** This figure presents a histogram of the distribution of sequencing count for OTU Finegoldia Magna from Romero et al. (2014) [15] dataset. The data are sparse with a right-skewed distribution.

To study and analyze microbiome data, metagenomic reads are processed for each microbiome sample to construct taxonomic profiles [16,17]. Then, the combination of the taxonomic profiles into one count table is called the Operational Taxonomic Unit (OTU) table, which is widely used in microbiome studies [18]. A typical OTU dataset contains measurements of abundance for OTUs, the total number of reads, and the number of samples. Table 1 shows what an OTU table looks like. The dimension of the table is $n \times m$, where $n$ denotes the number of metagenomic samples, and $m$ denotes the number of microbial features.

The entry $z_{ij}$ represents the number of reads from sample $i$ that mapped to microbial feature $j$, where $i \in [1, n], j \in [1, m]$. This number can be the abundance of taxa grouped at different levels such as species, genus, and family. $N_i, i \in [1, n]$ is the total number of sequence reads for sample $i$. Table 1 is a general representation of an OTU table.

**Table 1.** A general form of a microbiome OTU table.

| Sample/Species | OTU 1 | OTU 2 | OTU 3 | ... | OTU $m$ | Total Reads |
|---|---|---|---|---|---|---|
| Sample 1 | $z_{11}$ | $z_{12}$ | $z_{13}$ | ... | $z_{1m}$ | $N_1$ |
| Sample 2 | $z_{21}$ | $z_{22}$ | $z_{23}$ | ... | $z_{2m}$ | $N_2$ |
| ... | ... | ... | ... | ... | ... | ... |
| Sample $n$ | $z_{n1}$ | $z_{n2}$ | $z_{n3}$ | ... | $z_{nm}$ | $N_n$ |

*Differential Abundance and Normalization Methods for Microbiome Data*

Metagenomic samples may have different sequencing depths, so the metagenomic counts need to be normalized among samples [19]. Failure to normalize the metagenomic counts may increase the distribution bias and reduce the distribution power [20]. We summarize below three well-known normalization methods.

1. Scaling Methods. The idea of the scaling method is to divide the observed abundance $z_{ij}$ by a scaling (normalization) factor. More specifically, scaling is defined as follows.

$$\tilde{z}_{ij} = \frac{z_{ij}}{s_i},$$

where $\tilde{z}_{ij}$ is the normalized abundance for feature $j$ within sample $i$, and $s_i$ is the scaling (normalization) factor for sample $i$.

Some common scaling normalization methods include cumulative sum scaling (CSS) [21], median-of-ratios scaling factor (DESeq2) [22], analysis of composition of microbiomes (ANCOM) [23], and trimmed mean of M-values (TMM) [24].

2.  Log-ratio Methods. The most known log-ratio transformation used in microbiome data analyses is the centered log-ratio (clr) [25]. In particular, clr transforms the features by taking the log of the ratio between observed features and their geometric mean. Some common log-ratio normalization methods include centered log-ratio (CLR) transformation [26] and ALDEx [27].

3.  RNA-seq Methods. The RNA-seq methods are parametric methods. A large part of the variability in RNA-seq data arises from the sampling of the microbial ecosystem [28]. DESeq2 and edgeR are two popular methods from RNA-seq for testing differences across study groups. Both methods model the observed abundances using the negative binomial distribution. Some recent studies have indicated the poor performance of these two methods [20,29]. MetagenomeSeq is an alternative RNA-seq method. Instead of using a negative binomial model, MetagenomeSeq is based on a zero-inflated Gaussian (ZIG). For more details about a general zero-inflated model, please read Section 3.1.3. MetagenomeSeq has been applied to different microbiome studies and shows higher powers than most of the other differential abundance methods such as DESeq2 and edgeR [30,31].

## 3. Modeling Single Feature

Identifying algorithms remains a crucial part of recognizing patterns, regression, and classification. Thus, it is imperative to choose independent features [14]. In microbiome analysis, modeling single features can help identify a genome for microbes with low abundance and can reveal more taxonomic and functional information about specific members of the microbiome at the cellular level [32]. The biomarkers found can allow for low-abundance microbes to be analyzed in further detail, and provides information about microbiome interactions and individual microbes [14]. Two different applications related to gut and vaginal microbiomes are explained below.

The gut microbiome has been found to fluctuate due to the intensified medication used to treat flares of inflammatory bowel disease (IBD) [33]. By analyzing the 16S ribosomal RNA gene, Bacteroidetes and Firmicutes have been found to make up ninety percent of phylogenetic categories, giving these bacterial divisions a distinct role in the human gut microbiome [34]. As the importance of dominating bacterial divisions is discovered, it is equally as important to understand microbes with low abundance and their impact on IBD or other chronic diseases.

Lactobacilli are the most abundant vaginal bacteria in women [35]. Lactobacilli produce lactic acid, which acidifies the vagina to pH < 4 to restrict the growth of all bacteria and protect the vagina against pathogens. Lactobacilli also produce hydrogen peroxide to kill bacterial cells by destroying their cell walls [36]. The following are three questions that have arisen and need some answers: (1) What is the distribution of Lactobacilli species? (2) Given some covariates such as the subject's age and group, how can we model the Lactobacilli count? (3) Suppose we have longitudinal data with multiple time points, how can we identify the most significant time intervals for the Lactobacilli count?

To answer the above three questions, in general, there are three types of modeling of single microbiome features: (1) Probabilistic models for snapshot studies, where each subject provides only one sample; (2) longitudinal studies, which include multiple samples per subject over time; and (3) regression analysis. In this section, we focus on modeling a single microbiome feature, that is, $z_{ij}$ for a sample $i$ and feature $j$.

### 3.1. Probabilistic Models

The microbiome has been linked to some major human diseases such as obesity [37], diabetes [38], hepatic steatosis [39], inflammatory bowel diseases (IBD) [40], autism [41], food allergies [42], cardiovascular disease [43], depression [44], many types of cancer [45],

and more. Therefore, the human microbiome plays a vital role in the diagnosis, analysis, and treatment of these diseases [46]. For example, in order to determine if there is an association between a microbiome feature and the disease, we may need to detect the significance of the difference between the two groups. With appropriate probabilistic models identified successfully, we can improve the power of the test significantly [17].

Below is a list of the most common probabilistic models used in snapshot microbiome studies.

### 3.1.1. Poisson Model

Poisson distribution is one of the most common models used for modeling nonnegative count data. If a random feature count $Z_{ij}$ follows a Poisson distribution with mean $\lambda > 0$, then the probability mass function (pmf) is given by

$$P(Z_{ij} = k) = e^{-\lambda} \frac{\lambda^k}{k!}$$

for $k = 0, 1, 2, \ldots$. The parameter $\lambda$ is equal to the expected value of $Z_{ij}$ and also to its variance. This restriction is usually not true in most microbiome applications. Most often, the observed variation is greater than the mean so an extension to the Poisson model is more appropriate.

### 3.1.2. Negative Binomial Model

The negative binomial (NB) distribution is another probabilistic model for count data. It is especially useful when the sample variance exceeds the sample mean, known as over-dispersion.

Given a sequence of independent Bernoulli trials, each trial has two potential outcomes called "success" and "failure." In each trial, the probability of success is $p$ and failure is $1 - p$. We observe this sequence until a predefined number $r$ of successes occurs. Then the random number of observed failures, $Z_{ij}$ before the $r^{th}$ success is called a Negative binomial (NB) distribution, and its pmf is given by

$$P(Z_{ij} = k) = \binom{k + r - 1}{k} p^r (1 - p)^k,$$

where $r > 0$ and $0 \leqslant p \leqslant 1$ are two distribution parameters, and $k = 0, 1, 2, \ldots$.

### 3.1.3. Zero-Inflated Models

Although the negative binomial distribution is able to address the over-dispersion where the variance is greater than the mean, it is not appropriate for modeling sparse data with a high proportion of zeros. In order to handle this issue, zero-inflated and hurdle models are used to model read counts that have an excess of zeros.

A zero-inflated model is a mixture of two statistical processes; one always generates zero counts and the other generates both zero and nonzero counts [47].

As a result, the combined probability under a zero-inflated model is

$$P_{ZI}(Z_{ij} = k) = \phi \mathbf{1}_{\{k=0\}} + (1 - \phi) P(Z_{ij} = k), \tag{1}$$

where $\phi > 0$ is the probability of extra zeros. $P(Z_{ij} = k)$ stands for the probability determined by any baseline distribution such as Poisson, negative binomial, normal, or other parametric distributions. The corresponding distributions are known as zero-inflated Poisson (ZIP), zero-inflated negative binomial (ZINB), zero-inflated Gaussian (ZIG) distributions, etc.

### 3.1.4. Hurdle Models

Hurdle models, also known as zero-altered models, provide another way of dealing with the excess zeros in OTU counts [14,48]. A hurdle model consists of two components,

one generating the zeros and one generating the positive values. Unlike the zero-inflated model, the zero and non-zero counts are separated in the hurdle model.

The hurdle model is defined as

$$P_{\text{ZA}}(Z_{ij} = k) = \phi\mathbf{1}_{\{k=0\}} + (1 - \phi)P_{\text{tr}}(Z_{ij} = k), \tag{2}$$

where $P_{\text{tr}}(Z_{ij} = k)$ is a truncated version of $P(Z_{ij} = k)$.

$$P_{\text{tr}}(Z_{ij} = k) = \begin{cases} 0 & \text{if } k = 0 \\ P(Z_{ij} = k)/[1 - P(Z_{ij} = 0)] & \text{if } k > 0. \end{cases} \tag{3}$$

For example, if $P(Z_{ij} = k)$ comes from a negative binomial distribution, then $P_{\text{tr}}(Z_{ij} = k)$ is known as a zero-truncated negative binomial distribution.

Probabilistic models used in a snapshot microbiome is a growing research area. For example, Aldirawi et al. (2019) developed a statistical method for identifying the most appropriate probabilistic models for some discrete distributions with applications to microbiome data [17]. They have adjusted the Kolmogorov–Smirnov test (KS-test) to fit discrete probabilistic models with unknown parameters. Their developed approach can be applied to a general class of zero-inflated and hurdle models, then the estimated parameter can be calculated. Their method was applied to datasets related to lung and skin microbiomes. They found that beta binomial, beta negative binomial, and the corresponding zero-inflated and hurdle models are more appropriate compared to the commonly used discrete distributions such as Poisson, negative binomial, and the corresponding zero-inflated and hurdle models [49].

In order to test whether a specific feature follows any specific discrete or continuous distribution such as an exponential or zero-inflated negative binomial, there are some R packages available from the Comprehensive R Archive Network (CRAN, https://cran.r-project.org/, accessed on 15 December 2022). The most recent R package is "AZIAD" [50], which covers 27 discrete and continuous distributions. The AZIAD package provides maximum likelihood estimates for model parameters, likelihood ratio tests (LRT) for model selection, Kolmogorov–Smirnov tests (KS tests), the Fisher information matrix, and confidence intervals for parameter estimates.

### 3.2. Regression Analysis

In the previous section, we discussed the probabilistic models used for snapshot microbiome studies without covariates. Now, suppose the response variable is the number of counts of any specific feature, and there are some given covariates such as subject group. This is a regression problem. In this section, we discuss some of the well-known regression analysis models for microbiome data.

### 3.2.1. Generalized Linear Models

Generalized linear models (GLM), such as Poisson and negative binomial (NB) models, can be applied to count data [51]. The Poisson model is one of the most popular regression models for count data. The Poisson distribution is as fundamental to the analysis of count data as the normal is to continuous responses [52]. It has the simple probability mass function [51]:

$$P(Y_i = y \mid X_i) = \frac{\exp(-\mu_i)\mu_i^y}{y!}.$$

The Poisson model assumes that the number of read counts $Y_i$ is sampled from a Poisson distribution. The dependence of $\mu_i = E(Y_i)$ on the covariate vector $\mathbf{X}_i$ is usually written in the logarithmic form

$$\log \mu_i = \eta_i = \boldsymbol{\beta}^T\mathbf{X}_i; \quad i = 1, \dots, n.$$

Note that the variance of the Poisson model is equal to the mean.

Negative binomial (NB) regression is used for modeling count variables, usually for over-dispersed count outcome variables. This suggests it might serve as a useful approximation for modeling counts with variability different from its mean. The variance of a negative binomial distribution is a function of its mean and has an additional parameter, $\theta$, called the dispersion parameter. Suppose a random variable $Y$ is counted from a negative binomial distribution, then the variance of $Y$ is

$$\text{var}(Y) = \mu + \mu^2/\theta.$$

As the dispersion parameter becomes larger and larger, the negative binomial turns into a Poisson distribution.

The NB model is given with the following density function:

$$P(Y_i \mid X_i) = \frac{\Gamma(Y_i + \theta)}{\Gamma(Y_i + 1)\Gamma(\theta)} \left( \frac{\theta}{\theta + \mu_i} \right)^{\theta} \left( \frac{\mu_i}{\theta + \mu_i} \right)^{Y_i}.$$

The generalized linear model is based on the exponential family of distribution and unifies linear and nonlinear regression models. To use the GLM, it assumes that the distribution of the study variable is a member of the exponential family of distribution.

### 3.2.2. Vector Generalized Linear Models

When the count data are sparse with a significant percentage of zeros, GLM is not recommended because the proportion of zeros ($\phi_i$) must be linked to some distributions [53]. Although GLMs have been widely used, they have largely been confined to single parameter distributions belonging to the exponential family. Since there are many situations where the distribution is not a member of the exponential family, we need more flexible models than GLMs.

Yee (2015) [54] described a larger and more flexible statistical framework to extend GLMs, called Vector Generalized Linear Models (VGLMs). To fit a regression model with parameters $\theta_j$'s, VGLMs model each parameter as a linear combination of the explanatory variables after a (monotone) transformation. That is,

$$g_j(\theta_j) = \eta_j = \boldsymbol{\beta}_j^T \boldsymbol{x} = \beta_{(j)1}x_1 + \cdots + \beta_{(j)p}x_p, \quad j = 1, \ldots, M,$$

where $g_j$ is a parameter link function such as a logarithm or logit. Note that potentially every parameter is modeled using all explanatory variables $x_k$ and the parameters need not be a mean such as for GLMs.

Aldirawi (2020) extended VGLMs and modeled zero-inflated and hurdle regression models as follows [55]:

$$g(\phi_i) = \mathbf{G}_i^T \boldsymbol{\gamma}, \quad i = 1, \ldots, n$$
$$h_j(\theta_{ij}) = \mathbf{B}_{ij}^T \boldsymbol{\beta}_j, \quad i = 1, \ldots, n; j = 1, \ldots, b,$$

where $g$ and $h_1, \ldots, h_b$ are known link functions, $\boldsymbol{\gamma}, \boldsymbol{\beta}_1, \ldots, \boldsymbol{\beta}_b$ are regression coefficients, $\mathbf{G}_i = (r_1(\mathbf{x}_i), \ldots, r_s(\mathbf{x}_i))^T \in \mathbb{R}^s$ and $\mathbf{B}_{ij} = (q_{j1}(\mathbf{x}_i), \ldots, q_{jt_j}(\mathbf{x}_i))^T \in \mathbb{R}^{t_j}$ are the corresponding predictors, $r_i$'s and $q_{ji}$'s are known functions.

Examples include $\mathbf{G}_i = \mathbf{B}_{ij} = (1, x_{i1}, \ldots, x_{id})^T$ for a main-effects model and $\mathbf{G}_i = \mathbf{B}_{ij} = (1, x_{i1}, \ldots, x_{id}, x_{i1}x_{i2}, \ldots, x_{i,d-1}x_{id})^T$ for a model with both main effects and order-2 interactions.

Zero-inflated and hurdle regression models are widely used for modeling microbiome data. For example, Hu et al. (2018) introduced a zero-inflated beta-binomial (ZIBB) regression model to model the distribution of microbiome count data and to determine the association with a continuous or categorical phenotype of interest [56]. They found that their proposed ZIBB framework performs well in real data analysis and simulation studies. The proposed ZIBB method effectively controls type I errors and has higher power

than BBSeq, ZINB, and edgeR. An R package, ZIBBSeqDiscovery, is available on R CRAN. Xu et al.'s (2015) [57] study on modeling the gut microbiome of 400 independent subjects compared the performance of different methods for modeling microbiome data. These methods include some standard parametric and non-parametric models, zero-inflated models, and hurdle models. They compared some criteria such as the power, type I error, goodness of fit, and efficiency of parameter estimation. They used the Akaike information criterion (AIC) for model selection. Based on a real application of microbiome data, their method showed that zero-inflated and hurdle models have higher power, better-controlled type I errors, better goodness-of-fit, and more accurate parameter estimation. In addition, they found that the zero-inflated and hurdle models have some similar results in terms of goodness-of-fit and parameter estimation. Van den Elskamp et al. (2009) [58] discussed the statistical distributions used for modeling lesion counts in patients with multiple sclerosis (MS). The AIC model selection criteria on six different models showed that the negative binomial distribution provided the most optimal fit, followed by the Poisson-Inverse Gaussian and Poisson-Lognormal distributions.

### 3.2.3. Bayesian Models

Bayesian Models have been widely used for modeling microbiome data. As microbiome data are high dimensional and sparse in general, sometimes the GLM and VGLM models do not fit the data very well. To overcome these challenges, some Bayesian models were proposed. For example, Wadsworth et al. (2017) [59] proposed a Dirichlet-multinomial Bayesian variable selection (DMBVS) model that uses spike-and-slab priors for the selection of significance between covariates and microbiome feature. They applied the proposed model to both simulated data and publicly available data. The results showed the connection between a specific microbiome feature and particular metabolic pathways. Koslovsky et al. (2020) [60] proposed a Dirichlet-multinomial linear model with Bayesian variable selection (DMLMbvs) using spike-and-slab priors. Their approach can handle high-dimensional compositional data as well as clinical data. In addition, it can accommodate taxa heterogeneity when predicting phenotypic responses.

### 3.3. Longitudinal Microbiome Data

Longitudinal studies of the microbiome can uncover information involving the contributing factors, microbe interactions, and long-term effects of various health concerns. Recent studies have branched out into child development and factors of variation in adults with and without chronic diseases [61]. When studying the microbiome, longitudinal studies have proven to be useful. Observing individual features of a microbiome over a period of time yields information that could be important when discussing the interactions, or how the microbiomes interact with each other. Since these communities are continuously fluctuating and evolving, the time taken to conduct the study can be useful when comparing factors such as weather, disease progression, and the consistency of microbiomes [62].

Modeling sparse longitudinal microbiome data is challenging for a few reasons. First, the microbiome data are non-normally distributed. Therefore, methods with normal distributional assumption are not expected to perform well [63]. Second, microbiome data are sparse with a large proportion of zeros, which causes heterogeneity issue in the data. Third, longitudinal studies in general suffer from all forms of variability such as a different number of samples per subject, a different number of subjects per group, and samples not collected at consistent time points. Fourth, the repeated measurements in longitudinal data are correlated; therefore, taking into account the correlations among repeated measurements is necessary. Based on the above limitations, univariate tests (such as the t-test) and standard longitudinal models such as the generalized estimating equations (GEEs) are not recommended.

Below is a list of some common longitudinal models used in snapshot microbiome studies.

1. MetaDprof [64] is a smoothing spline-based method, and a well-known method for modeling longitudinal data [65,66]. MetaDprof is used for detecting differentially abundant features from metagenomic samples by comparing different conditions across time. There is a major limitation of the MetaDprof method. It assumes consistency in longitudinal microbial samples. For example, the same number of subjects per phenotypic group, the same number of samples from each subject, and the same time points [62].

2. MetaLonDA [62] is an R package that is capable of identifying significant time intervals of differentially abundant microbial features. It can be applied to any longitudinal count data such as metagenomic sequencing, 16S rRNA gene sequencing, or RNAseq. MetaLonDA relies on two modeling components. The NB distribution for modeling the features reads counts and the semi-parametric SS-ANOVA technique for modeling longitudinal profiles associated with different phenotypes. MetaLonDa is able to handle the metaDprof limitations. For example, it does not require the same number of subjects per group. The elapsed time between adjacent time points is flexible. One limitation of MetaLonDA is that when samples are sparse over time intervals, the fitted smoothing spline has a large variation.

3. Zero-inflated Beta regression model with random effects (ZIBR) model [63]. Chen and Li (2016) proposed a two-part zero-inflated beta regression model with random effects (ZIBR) for testing the association between microbial abundance and clinical covariates for longitudinal microbiome data. The proposed model includes a beta regression component to model non-zero microbial abundance, and a logistic regression component to model the presence/absence of a microbe in the samples. Each component includes a random effect to account for the correlations among the repeated measurements on the same subject. Based on a real microbiome data application, the ZIBR model performed better than the commonly used models such as binomial, zero-inflated Poisson, and negative binomial regression models.

4. Zero-inflated negative binomial mixed-effects (ZINBMM) model [15]. Romero et al. (2014) proposed a longitudinal vaginal microbiome study for comparing the vaginal microbiome feature (Lactobacillus) between two groups of women (pregnant and non-pregnant women). The zero-inflated negative binomial mixed-effects (ZINBLME) model was applied to model the read counts on the pregnancy status. In addition, negative binomial linear mixed effects (NBLME) and Poisson linear mixed effects (PLME) models were used in the model comparison. Based on their proposed method, the ZINBLME model provided the best fit based on AIC values. One limitation of Romero et al.'s (2014) model is that it can only be applied to count data [63].

5. Long Short Term Memory Networks (LSTM) [67]. Recently, Sharma and Xu (2021) proposed a deep learning framework for the feature extraction and analysis of temporal dependency in longitudinal microbiome sequencing data along with the host's environmental factors for disease prediction. The proposed methodology and an extensive analysis and comparison were applied to 100 simulated datasets across multiple time points and were applied to two real longitudinal human microbiome studies. The analysis showed that the proposed model significantly improves predictive accuracy.

## 4. Multivariate Microbiome Analysis

The structure and function of microbiomes are influenced by the interactions between microbes and the interaction between the microbe and some other factors such as host, drug, and environment. Those interactions have implications for the progression of diseases and clinical outcomes. Understanding these microbial communities and interactions is important for the recognition of microbiome association with host health, development, dysbiosis, and polymicrobial infections [1]. A microbiome can form a complex network of interacting bacteria, archaea, and fungi. Therefore, to understand the interactions between these microbes with a justified method, we have to be careful about the compositional

nature of such data, not having enough samples with respect to the number of features, and a lack of microbiome networks with known interactions.

### 4.1. Microbiome–Microbiome Interaction

Microbiome–microbiome interactions require knowledge of microorganism taxa, environment, and interactions with other microorganisms to understand the functionality of the microbiome [5]. One of the most common microbiome–microbiome interactions is the transference of molecular and genetic information between microorganisms [5]. This method of communication within microbiomes is yet to be completely understood, but knowledge of regulation, maintenance, and communication within these microbial systems can advance research in understanding pathogen development, antimicrobial drugs, and human health [68]. Further, certain microbiome–microbiome interactions cause imbalances in the microbiome which have implications for certain diseases. Therefore, analyzing the data of these microbiome interactions can contribute to therapies for certain diseases caused by imbalance, and disease prediction [69]. The following is a list of modeling techniques that can be used to study the microbiome–microbiome interactions.

1.  Bayesian Network (BNs): BNs are directed probabilistic graphical models that represent a probabilistic relationship between multiple species via a directed acyclic graph. The nodes in BN correspond to random variables, and the directed edges correspond to conditional dependencies between them. The absence of an edge connecting two nodes indicates independence or conditional independence between them. The Bayesian network is an appropriate tool for modeling the interactions of many microbial taxa. It has been used in microbiome studies. For example, Bennett's (2016) [70] study analysis is based on the construction of a Bayesian network using Dirichlet distributions to model the conjugate probabilities of the most common bacterial constituents in a stool sample. The results indicate that the Bayesian network adjusts the prior bacterial population distribution to more accurately reflect the transcriptionally active bacterial population.

2.  Graphical Gaussian models (GGMs) are undirected probabilistic graphical models that identify the conditional independence relations among the nodes, where the nodes correspond to multivariate normal distributed variables, and edges between these variables represent conditional dependencies. Zhao and Duan (2019) used GGM to learn the gene interactions in 15 specific types of human cancer [71]. The networks reveal conditional dependencies among the genes, and the weights of edges indicate the strength of the dependencies. The GGM networks reveal stable conditional dependences among the genes and confirm the essential roles played by the genes that encode proteins involved in the two key signaling pathways—PI3K/AKT/mTOR and Ras/Raf/MEK/ERK—in human carcinogenesis.

3.  SparCC: Sparse Correlations for Compositional data (SparCC) was developed by Friedman and Alm (2012) [72]. The method is capable of estimating correlation values from compositional data. SparCC estimates the linear Pearson correlations between the log-transformed components. Since these correlations cannot be computed exactly, SparCC utilizes an approximation that is based on the assumption that the number of OTUs is large and most OTUs are not strongly correlated with each other. In Friedman and Alm's (2012) [72] study, they infer a rich ecological network connecting hundreds of interacting species across 18 sites on the human body. SparCC shows that it can infer correlations with high accuracy even in the most challenging datasets.

4.  FastSpar was proposed recently by Watts et al. (2019) as a fast and parallelizable implementation of the SparCC algorithm with an unbiased P-value estimator [73]. One drawback of SparCC is the overestimated and biased p-value in some cases [74]. FastSpar produces equivalent OTU correlations as SparCC while greatly reducing run time, handling large datasets, and more accurate p-values. FastSpar has been used recently for modeling microbiome data. For example, Qiu et al. (2022) applied the FastSpar algorithm to analyze the soil and plant rhizosphere microbiome of

cotton plants in the presence of some cotton-specific fungal pathogen [75]. Their statistical analysis found that Fusarium oxysporum f.sp. vasinfectum (FOV) directly and consistently changed the rhizosphere microbiome. However, the biocontrol agents enabled microbial assemblages to resist pathogenic stress. Their study is essential for understanding core microbiome responses and the existence of plant pathobiomes, which provides an excellent framework for better plant disease management.

5.  SPIEC-EASI: SParse InversE Covariance Estimation for Ecological Association Inference (SPIEC-EASI) was proposed by Kurtz et al. (2015). It relies on algorithms for sparse neighborhood and inverse covariance selection [76]. It can handle some technical challenges related to microbiome data analysis. For example, the abundances of OTUs are compositional (Because the Counts are normalized). Thus, microbial abundances are not independent, and traditional statistical metrics such as the correlation-based methods for the detection of OTU-OTU relationships can lead to misleading results. Moreover, microbiome data are high dimensional data in general (the number of OTUs ($p$) is greater than the number of samples $n$); thus, inference of OTU-OTU association networks is required for an accurate inference. SPIEC-EASI can address both of these issues. Kurtz et al.'s (2015) application to gut microbiome data using SPIEC-EASI produced more consistent and sparser interaction networks than SparCC and CCREPE [76].

6.  CCLasso: Correlation inference for Compositional data through Lasso (CCLasso) is Similar to SparCC. CCLasso explicitly considers the compositional nature of the metagenomic data in correlation analysis, and it has the advantage that the estimated correlation matrix for compositional data is positive definite [77]. The performance of CCLasso is compared with SparCC through some simulation studies and a real microbiome example from the Human Microbiome Project (HMP). The results show that CCLasso gives a more accurate estimation for the correlation matrix than SparCC as well as better edge recovery.

7.  Relevance Networks (RN): Relevance networks is an unsupervised learning methodology used in functional genomics and microbiome data with the principal advantages being the ability to (1) include features of more than one data type, (2) represent multiple connections between features, (3) capture both negative and positive correlations, and (4) handle missing data [78]. In the RN method, each set of $p$ edges completely connects the $n$ nodes, and each pair of nodes is connected by a single edge with a score. A study by Werhli et al. (2006) [79] compared three different modeling and inference paradigms, relevance networks (RNs), graphical Gaussian models (GGMs), and Bayesian networks (BNs). The result shows that on Gaussian observational data, BNs and GGMs were found to outperform RNs. There was not a significant difference between BNs and GGMs on observational data in general. However, for interventional data, BNs outperform GGMs and RNs.

8.  Local Similarity Analysis (LSA): There are many techniques for identifying the relationship between species and associations between species and environmental factors such as Pearson Correlation Coefficient (PCC), and canonical correlation analysis (CCA) analysis. LSA is a novel technique that can identify more complex dependence associations among species as well as associations between species and environmental factors without requiring significant data reduction [80]. Based on a marine microbial observatory dataset application, LSA identified unique, significant associations that were not detected by PCC analysis. LSA can be extended for time series data with replicates.

### 4.2. Host/Drug–Microbiome Interaction

The microbiome can impact the host depending on the condition and environment. Drug–microbiome interactions focus on how different drugs affect microbiomes, and host–microbiome interactions focus on how microbiomes affect the host [5,81]. Drug–microbiome interactions indicate that the effect of drugs on microbiomes depicts disturbances and

functional alterations. Antibiotics have been found to disrupt the microbial balance, creating resistant bacteria which can make future treatments more difficult to deal with because of antibiotic resistance [6]. Further, non-antibiotic drugs have been found to change the composition and function of the microbiome [81]. Microbiome interaction with the host begins with identifying what type of microbiome is affecting the host, and how beneficial or harmful it is to the host [5].

Understanding what kind of changes can influence the microbiome, and how, could lead to further studies on ways to make the microbiome stronger by improving the effects of certain treatments [5]. Therefore, modeling the host/drug–microbiome interaction is important for understanding the significance of the interactions and implications for different drugs and microorganism taxa. For example, Maier et al. (2018) [82] discussed the extensive effects of non-antibiotic drugs on the gut microbiome. In their study, more than 1000 marketed drugs were screened against 40 representative gut bacterial strains, and they found that 24% of the drugs with human targets inhibited the growth of at least one strain. Particular classes, such as the chemically diverse antipsychotics, were overrepresented in this group. The effects of human-targeted drugs on gut bacteria are reflected by their antibiotic-like side effects in humans. Therefore, the study explains the necessity of accounting for potential medication-related confounding effects in future microbiome disease association studies.

### 4.3. Multivariate Longitudinal Data

Multivariate longitudinal data analysis provides significantly more information on the dynamics of the microbiome interaction networks than univariate longitudinal methods. It is very important to understand the relationships among taxa over time. These relationships can have a positive, negative, or no impact on the taxa involved. In this section, we review two strategies to identify the associations between longitudinal microbiome data.

1.  Dynamic Bayesian Network: A Dynamic Bayesian Network (DBN) is "a Bayesian network extended with additional mechanisms that are capable of modeling influences over time" [83]. DBN has been used recently for modeling multiple features jointly for longitudinal data. For example, Lugo-Martinez (2019) [84] proposed a study based on DBN for analyzing longitudinal microbiome data. They applied their approach to three different microbiome datasets including infant gut, vaginal, and oral cavity microbiomes. The results provide evidence that microbiome alignments coupled with DBN improve predictive performance over previous methods and enhance our ability to infer biological relationships within the microbiome and between taxa and clinical factors. In McGeachie et al.'s (2016) [85] study, DBN was applied to longitudinal infant gut microbiomes and the predictive performance was analyzed. The DBN model explicitly captured specific relationships and general trends in the data by increasing amounts of Clostridia, residual amounts of Bacilli, and increasing amounts of Gammaproteobacteria. The prediction performance of DBNs with fewer edges was accurate. DBN provided quantitative likelihood estimates for rare abruptions events. DBN was able to identify important relationships between microbiome taxa and predict future changes in microbiome composition.

2.  Multivariate Granger causality. The Granger causality network model was proposed by Granger (1969) [86], which was originally developed for economics but has now been used extensively in neuroscience and microbiome data analysis [87]. Variable $X$ is the "Granger cause" of variable $Y$ if the histories $X$ and $Y$ together predict the current value of $Y$ better than the history of $Y$ alone [88]. Several multivariate extensions of Granger causality have been developed recently [89–93]. For example, Mainali et al. (2019) [92] show the superiority of multivariate Granger causality over the traditional correlation methods, showing a weak negative relationship between correlation and causality, and a strong positive relationship, whereas almost all strong negative interactions. One limitation of this method is that it does not take into

consideration the clinical or demographic variables when building the interaction network [94].

### 4.4. Multivariate Regression Analysis

In Section 3.2, we review the regression models that model each feature independently by modeling the mean counts or some transformation of the counts via a link function. In this section, we discuss some regression analysis methods that focus on modeling OTUs jointly (i.e, in a multivariate count fashion). Below is a review of the most common multivariate regression models.

1.  Zero-inflated generalized Dirichlet multinomial (ZIGDM) model [95]. The ZIGDM is proposed for modeling multivariate taxon counts. The ZIGDM regression model was proposed to link microbial abundances to covariates and develop a fast expectation–maximization (EM) algorithm to efficiently estimate the parameters. Based on some simulation studies and an application related to the gut microbiome dataset, the ZIGDM test is more powerful at detecting differential mean/dispersion and is more robust to the underlying distribution if the counts are zero-inflated. If the taxon counts are not zero-inflated, the generalized Dirichlet multinomial (GDM) tests are more desirable. In addition, the GDM provides a superior fit to taxon counts compared to the Dirichlet multinomial (DM), and the ZIGDM can further improve the goodness-of-fit for taxa with many zero counts.

2.  Bayesian nonparametric multivariate negative binomial regression with zero-inflation (BNP-ZIMNR) model [96]. BNP-ZIMNR is used to analyze multivariate count responses of microbiome data. Zero-inflated negative binomial (ZINB) distribution is used for modeling OTU counts under the assumption that OTU counts are either equal to zero or follow a negative binomial distribution. Nonparametric regression prior models were built on the probability of an OTU count being zero and the mean count of an OTU to study the effects of covariates on microbial communities. Based on some simulation studies and a real chronic wound microbiome dataset, the proposed BNP-ZIMNR model yields superior parameter estimates and model fit in various settings.

3.  Bayesian Dirichlet-multinomial (BDM) regression model [59]. The proposed model allows for the selection of significant associations between a set of covariates and microbiome features. The statistical inference is conducted through a Markov Chain Monte Carlo (MCMC) algorithm, and the selection of the significant covariates is based on posterior probabilities of inclusions and the thresholding of the Bayesian false discovery rate. The proposed model has been applied to simulated data and real microbiome applications. Compared to some other methods, the BDM model is more accurate and has the lowest false positive as well as false negative rates.

4.  Logistic Normal Multinomial (LNM) Regression Model [97]. In order to select the covariates and estimate the corresponding regression coefficients, a penalized likelihood estimation method was developed for variable selection and estimation. The Monte Carlo Expectation-Maximization algorithm was applied to implement the penalized likelihood estimation. Compared to the commonly used Dirichlet-multinomial regression model for count data, the LNM model provides a more flexible way of modeling the dependency of the bacterial composition.

5.  Dirichlet-multinomial (DM) regression model [98]. Because microbiome data are high dimensional data, a penalized likelihood approach was developed to estimate the regression parameters and to select the variables by imposing a sparse group $l_1$ penalty to encourage both group-level and within-group sparsity. A variable selection procedure and an efficient block-coordinate algorithm were developed to solve the optimization problem. Based on some extensive simulations and a real application related to the human gut microbiome, the sparse DM regression can result in better identification of the microbiome-associated covariates than models that ignore overdispersion.

## 5. Conclusions

Sparsity, skewness, and high dimensionality are some of the main challenges for microbiome data analysis and have drawn considerable attention. Biases and lack of powers may be introduced if the excessive zeros observed in the data are handled inappropriately.

In this paper, we discussed the microbiome data challenges and the data representation. Then, we reviewed two types of statistical analysis of microbiome data; first, modeling Univariate OTU or features separately and independently, and second, modeling multiple OTUs or features simultaneously.

For the first type of analysis, we reviewed three types of analysis: (1) probabilistic models (without covariates); (2) modeling longitudinal microbiome data where there are multiple time points for each subject; and (3) regression analysis where the response variable is the OTU count. The second type of analysis is based on modeling multiple OTUs or features simultaneously. This kind of analysis may foster our understanding of interactions between species, or building a network among species. We reviewed four types of multivariate features: (1) microbiome–microbiome interactions; (2) host/drug–microbiome interactions; (3) longitudinal data; and (4) regression analysis.

**Author Contributions:** Conceptualization, H.A.; methodology, H.A. and F.G.M.; software, H.A.; validation, H.A. and F.G.M.; formal analysis, H.A.; investigation, H.A. and F.G.M.; writing—original draft preparation, H.A. and F.G.M.; writing—review and editing, H.A. and F.G.M.; project administration, H.A.; funding acquisition, H.A. All authors have read and agreed to the published version of the manuscript.

**Funding:** This research was partially supported by CSUSB 2022 Summer Research Grant.

**Data Availability Statement:** Not applicable.

**Conflicts of Interest:** The authors declare no conflict of interest.

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
