# Peer review of "Univariate and Multivariate Statistical Analysis of Microbiome Data: An Overview"

_2673-8007, doi:10.3390/applmicrobiol3020023_

Round 1

Reviewer 1 Report

In this manuscript authors discuss the challenges associated with microbiome data and its data representation. Further, they also review two types of statistical analysis of microbiome data. While the manuscript is well written, it can be organized in a better manner for clarity. Some points of consideration are:

Major points:

1.      On line 101, authors mention about some normalization methods such as CLR, CSS etc. It would be worth discussing other popular approaches such as metagenomeSeq, a zero-inflated beta model or ZIBSeq. These approaches can be put together in one subsection mentioning “Differential abundance analysis” for clarity.

2.      While authors mention inclusion of covariates along with OTU data, some popular methods for studying the association between the microbiome and covariates seem to have been missed out. Worth mentioning, Dirichlet-multinomial Bayesian variable selection or DMBVS, a Bayesian zero-inflated negative binomial (ZINB) also referred to as Integrative Bayes, and a Dirichlet-multinomial linear model with Bayesian variable selection or DMLMbvs.

3.      In Lines 60-66, authors mention about microbial analysis and its impact on areas beyond human health. However, they provide just one use case “analyze the water of hydraulic fracturing” which looks a bit isolated. If mentioning the same, they should also provide a comprehensive overview of other examples pertaining to impact of microbiome beyond human health.

4.      In the Multivariate Longitudinal Data section, authors touch upon use of LSTM for longitudinal microbiome data. However, it will be good to include a paragraph  in the discussion/ conclusion section mentioning how the microbiome data analysis field is evolving with the use of neural networks and deep learning. Appropriate references such as ”MetaNN Lo et al.” and “taxoNN, Sharma et al.” should be cited accordingly.

Minor points:

Paper has a few grammatical errors that need to be corrected. A thorough read of the paper is necessary to go through all typos. Some errors include:

1.      In Line 54: word being missing: Correct line “instead of interpreted by humans” to “instead of  being interpreted by humans”

2.      In line 73: “Host” should be in small letters

3.      Line 84: OTUs mentioned here but full form mentioned later

4.      In Line 328: Correct spelling of Microbiome

5.      In Line 558: Correct “analyst” to analysis

Author Response

  1.     On line 101, authors mention about some normalization methods such as CLR, CSS etc. It would be worth discussing other popular approaches such as metagenomeSeq, a zero-inflated beta model or ZIBSeq. These approaches can be put together in one subsection mentioning “Differential abundance analysis” for clarity. 

Thank you. We added a subsection “2.1: Differential Abundance and Normalization Methods for Microbiome Data”. 

  1.     While authors mention inclusion of covariates along with OTU data, some popular methods for studying the association between the microbiome and covariates seem to have been missed out. Worth mentioning, Dirichlet-multinomial Bayesian variable selection or DMBVS, a Bayesian zero-inflated negative binomial (ZINB) also referred to as Integrative Bayes, and a Dirichlet-multinomial linear model with Bayesian variable selection or DMLMbvs.

Thank you for the suggestion. We added a new subsection “3.2.3. Bayesian Models” and discussed some of the Bayesian approaches. 

  1.     In Lines 60-66, authors mention about microbial analysis and its impact on areas beyond human health. However, they provide just one use case “analyze the water of hydraulic fracturing” which looks a bit isolated. If mentioning the same, they should also provide a comprehensive overview of other examples pertaining to impact of microbiome beyond human health.

Thank you for this point. We added one more example. The implications of using fossil fuels have also been evident from a multi-omics study……..

  1.     In the Multivariate Longitudinal Data section, authors touch upon use of LSTM for longitudinal microbiome data. However, it will be good to include a paragraph  in the discussion/ conclusion section mentioning how the microbiome data analysis field is evolving with the use of neural networks and deep learning. Appropriate references such as ”MetaNN Lo et al.” and “taxoNN, Sharma et al.” should be cited accordingly.

Thanks for this point. Neural networks and deep learning are two large topics. A full paper is needed to discuss them. In this paper, we just mentioned a few examples related to Multivariate Longitudinal Data. In one of them, we discussed network analysis. 

Minor points:

Paper has a few grammatical errors that need to be corrected. A thorough read of the paper is necessary to go through all typos. Some errors include:

  1.     In Line 54: word being missing: Correct line “instead of interpreted by humans” to “instead of  being interpreted by humans”

Thank you. We added the word “being”. 

  1.     In line 73: “Host” should be in small letters

We edited it. 

  1.     Line 84: OTUs mentioned here but full form mentioned later

Thanks for the great comment. We added the “Operational Taxonomic Unit (OTU)” in its first appearance (line 84).

  1.     In Line 328: Correct spelling of Microbiome

Thank you. We edited it. 

  1.     In Line 558: Correct “analyst” to analysis

Thank you. We edited it. 

Reviewer 2 Report

The authors gave a very detailed and comprehensive review of statistical method used in microbiome data analysis. I only have a few minor suggestions and comments:

  • 1. This is another way to deal with zeros in microbiome data: imputation. For example: PMID: 34183041

2. Many methods are described in the paper. Does authors have any suggestions how to chose the best method?

3. More details about normalization methods?

Author Response

The authors gave a very detailed and comprehensive review of statistical method used in microbiome data analysis. I only have a few minor suggestions and comments:

  • 1. This is another way to deal with zeros in microbiome data: imputation. For example: PMID: 34183041

Thank you for the suggestion. We added a new subsection “3.2.3. Bayesian Models” and discussed some other ways related to Bayesian analysis.  

  1. Many methods are described in the paper. Does authors have any suggestions how to chose the best method?

We are not making blanket recommendations one way or the other, but simply want to encourage researchers and readers to carefully read and consider different methods. 

  1. More details about normalization methods?

Thank you for the great suggestion. We added a subsection “2.1: Differential Abundance and Normalization Methods for Microbiome Data”.

Reviewer 3 Report

This paper reviews some of the most important methods for modelling the microbiome data with single and multiple species. The authors highlight some of the main models for studying both microbiome-microbiome interaction and host/drug-microbiome interaction. In addition, they discuss the main longitudinal models for multivariate count response, and review the regression analysis methods that focus on modeling multivariate count responses. 

1.     The first sentence of the abstract is questionable, “Microbiome data is a high dimensional, sparse, compositional, and over-dispersed.” The word “a” seems unnecessary.

2.     In the abstract, it is better to describe the main contributions of this paper, rather than only telling readers the contents of paper.

3.     Abbreviations, such as “Operational Taxonomic Unit (OTU)” should be described for its first appearance, rather than appeared in line 90. Other abbreviations include pmf in line 159

4.     In line 90-105, what are the differences between the normalization methods for metagenomic samples and those for RNA-seq data (RPKM, FPKM, TPM etc., see Wang, P.; Chen, S.; Yang, S. Recent Advances on Penalized Regression Models for Biological Data. Mathematics 2022, 10, 3695.).

5.     What is the difference between z_ij and Z_ij? As well as m and M?

6.     Punctuation marks should be added after some equations

7.     Blank at the beginning of the sentence is unnecessary in line 230.

8.     The paper mainly introduces some statistical models for microbiome data, it is better that if some main biological findings from those models are dicussed. 

Author Response

  1.     The first sentence of the abstract is questionable, “Microbiome data is a high dimensional, sparse, compositional, and over-dispersed.” The word “a” seems unnecessary.

Thanks for this notation. I just deleted the “a” letter.

  1.     In the abstract, it is better to describe the main contributions of this paper, rather than only telling readers the contents of paper.

Thank you. We edited the abstract based on your recommendation. 

  1.     Abbreviations, such as “Operational Taxonomic Unit (OTU)” should be described for its first appearance, rather than appeared in line 90. Other abbreviations include pmf in line 159

Thanks for the great comment. I just added the “Operational Taxonomic Unit (OTU)” in its first appearance (line 84). I defined the probability mass function (pmf) for the Poisson distribution in line 147, so I don’t have to describe it again in line 159. 

  1.     In line 90-105, what are the differences between the normalization methods for metagenomic samples and those for RNA-seq data (RPKM, FPKM, TPM etc., see Wang, P.; Chen, S.; Yang, S. Recent Advances on Penalized Regression Models for Biological Data. Mathematics 2022, 10, 3695.).

Great question. We added a subsection “2.1: Differential Abundance and Normalization Methods for Microbiome Data”. In this subsection, we explained the normalization methods further.

  1.     What is the difference between z_ij and Z_ij? As well as m and M?

$z_{ij}$ represents the number of reads from sample $i$ that mapped to microbial feature $j$. If you look at the OTU table, we use $z_{ij}$. However, $Z_{ij}$ is a random variable. When we use any distribution such as Poisson distribution, we use $Z_{ij}$. $m$ is the number of features. The letter $M$ is not included in the OTU table. It’s included in the regression part only, it represents the parameter. 

  1.     Punctuation marks should be added after some equations

Thank you. We edited this part. 

  1.     Blank at the beginning of the sentence is unnecessary in line 230.

Yes, we edited that. 

  1.     The paper mainly introduces some statistical models for microbiome data, it is better that if some main biological findings from those models are dicussed. 

Actually, we discussed most of the findings and we provided some examples. 

Round 2

Reviewer 1 Report

No further comments.